# Bioactivity Screening and Chemical Characterization of Biocompound from Endophytic *Neofusicoccum parvum* and *Buergenerula spartinae* Isolated from Mangrove Ecosystem

**DOI:** 10.3390/microorganisms11061599

**Published:** 2023-06-16

**Authors:** Rafael Dorighello Cadamuro, Isabela Maria Agustini da Silveira Bastos, Ana Claudia Oliveira de Freitas, Marilene da Silva Rosa, Geovanna de Oliveira Costa, Izabella Thaís da Silva, Diogo Robl, Patricia Hermes Stoco, Louis Pergaud Sandjo, Helen Treichel, Mário Steindel, Gislaine Fongaro

**Affiliations:** 1Department of Microbiology, Immunology and Parasitology, Federal University of Santa Catarina, Florianópolis 88040-900, SC, Brazil; cadamuro.rafael@gmail.com (R.D.C.); isabelamaria646@gmail.com (I.M.A.d.S.B.); anaclaudia.olifr@gmail.com (A.C.O.d.F.); marilene.rosaa@gmail.com (M.d.S.R.); izabella.thais@ufsc.br (I.T.d.S.); diogo.robl@ufsc.br (D.R.); patricia.stoco@ufsc.br (P.H.S.); mario.steindel@ufsc.br (M.S.); 2Department of Chemistry, Federal University of Santa Catarina, Florianópolis 88040-900, SC, Brazil; cogeovanna@gmail.com (G.d.O.C.); p.l.sandjo@ufsc.br (L.P.S.); 3Department of Pharmaceutical Sciences, Federal University of Santa Catarina, Florianópolis 88040-900, SC, Brazil; 4Laboratory of Microbiology and Bioprocess, Federal University of Fronteira Sul, Erechim 99700970, RS, Brazil; helen.treichel@uffs.edu.br

**Keywords:** endophytic fungi, bioprospection, microbial control, mangrove

## Abstract

The discovery of biomolecules has been the subject of extensive research for several years due to their potential to combat harmful pathogens that can lead to environmental contamination and infections in both humans and animals. This study aimed to identify the chemical profile of endophytic fungi, namely *Neofusicoccum parvum* and *Buergenerula spartinae*, which were isolated from *Avecinnia schaueriana* and *Laguncularia racemosa*. We identified several HPLC-MS compounds, including Ethylidene-3,39-biplumbagin, Pestauvicolactone A, Phenylalanine, 2-Isopropylmalic acid, Fusaproliferin, Sespendole, Ansellone, Calanone derivative, Terpestacin, and others. Solid-state fermentation was conducted for 14–21 days, and methanol and dichloromethane extraction were performed to obtain a crude extract. The results of our cytotoxicity assay revealed a CC_50_ value > 500 μg/mL, while the virucide, Trypanosoma, leishmania, and yeast assay demonstrated no inhibition. Nevertheless, the bacteriostatic assay showed a 98% reduction in *Listeria monocytogenes* and *Escherichia coli*. Our findings suggest that these endophytic fungi species with distinct chemical profiles represent a promising niche for further exploring new biomolecules.

## 1. Introduction

Various pathogens have inflicted harm on humanity for many years, leading to sickness and fatalities. Viruses can be categorized into two primary groups: enveloped and non-enveloped viruses. Non-enveloped viruses are known for their resistance to various environmental conditions, including changes in pH, temperature, and exposure to ultraviolet radiation [1]. Non-enveloped viruses have been shown to exhibit excellent resistance to various environmental factors such as pH, temperature, and ultraviolet radiation. Among these non-enveloped viruses, some have been found to replicate precisely in enterocytes, a type of cell located in the epithelium of mammals responsible for nutrient and water absorption [1]. Enteric viruses employ the fecal–oral route, whereby they are shed in fecal matter and subsequently contaminate water bodies, posing a risk of infection to various organisms [2].

Chagas disease, another sickness caused by the *Trypanosoma cruzi* parasite, is a widespread health concern affecting approximately 6 million people globally, with an additional 70 million individuals living in high-risk areas [3,4,5]. Unfortunately, despite the development of two drugs back in the 1960s, only one of them is currently permitted for use in Brazil due to their limited efficacy and harmful side effects during the chronic phase of the illness [6,7].

The issue of bacterial infections and their resistance to antimicrobial agents has reached critical levels and poses a serious threat to public health. This problem is not confined to a single region but is pervasive globally. The root cause of this challenge lies in the natural evolution of these microorganisms, which undergo genetic changes that make them resistant to various antimicrobials. Both gram-negative and gram-positive bacteria contribute to this issue by serving as significant contaminants and causing a plethora of infections, underscoring the severity of this public health crisis [8,9].

Visceral leishmaniasis (VL) is a grave public health issue that disproportionately affects impoverished communities in Asia, East Africa, South America, and the Mediterranean region. Over 600 million individuals reside in high-risk regions for this disease, with an estimated 50,000 to 90,000 new cases arising each year. Shockingly, ten countries, namely Bangladesh, Brazil, China, Ethiopia, India, Kenya, Nepal, Somalia, South Sudan, and Sudan, account for more than 95% of reported VL cases [10].

In Latin America, Visceral leishmaniasis (VL) is a zoonotic disease caused by *Leishmania infantum* and is endemic in 13 countries. The disease is transmitted by the sand fly *Lutzomyia longipalpis*, with dogs being the primary carriers of infection. In 2019, Brazil reported a staggering 2529 new cases, accounting for 97% of all cases in Latin America. Alarming statistics reveal that 35% of those affected were children under the age of 10, and the disease’s lethality rate stood at 9%, making it one of the highest in the world [11]. All of these pathogens highlight the need to search for new bioactive compounds in different niches and microorganisms.

Mangrove ecosystems are intricate habitats consisting of forest formations distributed in transition zones between terrestrial areas and coastal regions of the ocean. These habitats pose significant challenges to organisms and microorganisms, such as low dissolved oxygen levels, high levels of organic matter, and elevated salinity. Moreover, the number of plant species capable of withstanding such harsh environmental conditions is limited [12].

Endophytic fungi have emerged as a promising target for discovering new compounds with potential activity against pathogens. In Brazil, three species are known, *Laguncularia racemosa, Avicennia schaueriana*, and *Rhizophora mangle* [13]. This is due to their genetic diversity and remarkable ability to synthesize diverse metabolites, making them a promising-to-be-fully-explored niche [14,15].

Endophytic fungi, intimately associated with plant tissues, establish unique ecological interactions that benefit the fungi and the host plant. These fungi can synthesize a diverse array of secondary metabolites, which can aid in the growth and development of the plant or provide defense against potential pathogens. Unlike plant pathogens, endophytic fungi do not harm plant tissue. Therefore, have become an essential focus of study for discovering new bio compounds with potential applications in medicine and agriculture [16,17,18]. Several related endophytic fungi from mangroves have been used to seek compounds against bacteria and viruses [19]. This study aimed to perform the isolation of endophytic fungi from mangrove plants, solid-state fermentation of axenic cultures for 14–21 days, chemical extraction, and assay several pathogens, such as: bacteria, yeasts, leishmania, Trypanosoma and viruses.

## 2. Material and Methods

### 2.1. Endophytic Fungi

The Mario Steindel Endophytic Fungi Collection is a biobank comprised of endophytic fungi that were isolated from mangrove plants (*Avecinnia schaueriana*, *Laguncularia racemosa*, and *Rhizophora mangle*) of region of Itacorubi and Ratones, Florianópolis, Santa Catarina.

Disinfection was performed at the University Federal of State of Santa Catarina according to protocol, using a rinse of ethanol, hypochlorite, and water; following the process of isolation, internal parts of samples were required to be cut and exposed to internal factors at potato-agar-dextrose (Kasvi) in Petri dishes and were incubated for 14 days at 28 °C [20].

The identification of the endophytic fungi isolated from leaves of *A. schaueriana* and *Laguncularia racemosa* demanded a thorough analysis of micromorphological characteristics, including the distribution of fragments and microculture, as well as macromorphological and molecular analysis of the ITS1 region of rDNA [21]. The genetic material was extracted using axenic cultures in Petri dishes and incubated for 14–30 days at 28 °C.

Two endophytic fungi were selected from the Mario Steindel Endophytic Fungi Collection, housed at the Federal University of Santa Catarina, which contains endophytic fungi isolated from the mangrove region of Itacorubi and Ratones (27°35′16.1″ S 48°3031.2″ W and 27°28′15.7″ S and 48°29′31.1″ W, Florianópolis-Santa Catarina). *Neofusicoccum parvum* and *Burgueneraria spartinae* were isolated from samples collected from *A. schaueriana* and *Buergenerula spartinae* from the *L. racemosa*, Itacorubi regions and were deposited in GenBank (OQ30047–*Neofusicoccum parvum*; OQ300436–*Buergenerula spartinae* [22,23].

### 2.2. Solid-State Fermentation

To prepare the inoculum of axenic cultures, fungal biomass was harvested from fungal colonies grown on PDA plate (14 days at 28 °C) with addition of 10 mL of Tween 80 (0.01% *w*/*w*) [23]. An inoculum containing 10^6^ spores was added to 90 g of commercial sterile organic rice and 90 mL of distilled water inside Erlenmeyer flasks. The fermentation process was carried out for 14–21 days to produce biocompounds, with a control group set up without spores [24].

### 2.3. Chemical Profile

After the indicated time of fermentation, the fungal biomass was subjected to chemical maceration using methanol-PA (Sigma, St. Louis, MO, USA) (CH_3_OH) and dichloromethane-PA (Sigma) (CH_2_Cl_2_) in proportion 1:1 (*v*/*v*) for four days, using simple filtration and a rotary evaporator, following solubilization at 800 µg/mL on acetonitrile (LCMS grade, Sigma-Aldrich, St. Louis, MO, USA) to analyses using ultra-performance liquid chromatography–mass spectrometry (Appendix A: Ultra-Performance Liquid Chromatography-Mass Spectrometry (UPLC-MS) of crude extract from the fungus *Buergunerula spartinae* after solid-state fermentation; Appendix A: Ultra-Performance Liquid Chromatography-Mass Spectrometry (UPLC-MS) of crude extract from the fungus *Neofusicoccum parvum* after solid-state fermentation). Chromatographic analyses were performed on an Acquity H-Class UPLC-PDA system (Waters Co., Milford, AS, USA). An Acquity UPLC BEH C18 (50 × 2.1 mm i.d., 1.7 µm) column (Waters Co., USA) was used for the analysis. All solutions prepared for the UPLC analyses were filtered through a 0.22 µm hydrophobic membrane made of cellulose. 

The column was maintained at 40 °C during the analyses. MS data were obtained using a quadrupole orthogonal acceleration time-of-flight (QTOF) mass spectrometer, Xevo GS-2 QTof, with an electrospray ionization (ESI) source, operating in both positive and negative modes, with the mass range between 100 and 1200 Da and a scan time of 1 s. The mobile phase system consisted of a gradient of 0.1% aqueous formic acid (pH 3.0) (A) and ACN (B) at a flow rate of 0.3 mL/min. The gradient consisted of 0–2 min, 90% A/10% B; 2–10 min, 55% A/45% B; 10–15 min, 10% A/90% B; 15–20 min, 90% A/10% B. The injection volume was 2 µL. The instrument settings in the positive mode were a capillary voltage of 3.0 kV, sampling cone voltage of 40 V, a source offset voltage of 80 V, desolvation temperature of 200 °C, source temperature of 80 °C, cone gas flow of 50 L/h, and desolvation gas flow of 500 L/h. Nitrogen was used as the nebulizer gas, and argon was the collision gas. MS and MSE data (in two scan functions) were acquired in the centroid mode and monitored with a scan time of 1 s. The collision energy was 6 eV in function one and ramped from 25 to 35 eV in function 2. To assure accurate mass values, data were corrected during the acquisition by an external reference (LockSprayTM) named leucine-enkephalin solution (1 ng/mL) at a flow rate of 20 µL/min. System control and data processing were performed using MassLynx 4.1 software (Waters Co., Augusta, GA, USA).

### 2.4. Obtation of Crude Extract

The fermented samples were subjected to chemical maceration by adding 50 mL of a 1:1 (*v*/*v*) mixture of dichloromethane (CH_2_Cl_2_–PA 90%) and methanol (CH_3_OH–PA 90%). After maceration, a simple filtration step was performed to remove cellular debris, followed by rotary evaporation using a pneumatic pump at maximum pressure (Bünchi–Vacuum Pump V-700, Sigma Aldrich, São Paulo, Brazil) to accelerate the evaporation of organic solvent residues in a volumetric flask. The temperature of the water bath was maintained between 30–55 °C. The resulting total extract contained both polar and non-polar compounds and was subsequently subjected to lyophilization for 24 h.

### 2.5. Biological Assays

Biological assays were conducted using the methanolic/dichloromethane crude extract, including cytotoxicity, virucidal, yeast, leishmania, Trypanosoma, and bacteriostatic assays.

The crude extract evaluated were solubilized in dimethyl sulfoxide (DMSO), resulting in a final concentration of 1% DMSO in cell culture assays, which is considered a safe concentration for cell culture experiments. The assays were performed using a concentration of 500–50 μg/mL.

#### 2.5.1. Cytotoxicity Assays

Cell viability assays are essential in determining the cytotoxicity of bioactive compounds. In this study, three different cell lines were utilized, including VERO (ATCC CCL-81TM) monkey kidney fibroblasts, L929 (ATCC CCL-185TM) rat fibroblasts, and A549 (ATCC CCL-1TM) human lung cancer cells. These cell lines were cultured using Dulbecco’s Modified Eagle Medium (DMEM; Thermo Fisher Scientific, Warsaw, Poland) supplemented with 10% fetal bovine serum (Thermo Fisher Scientific, Poland) incubated at 37 °C with 5% CO_2_. After 24 h of seeding the cells into 96-well plates at a density of 2.5 × 10^4^ cells per well, the cytotoxicity was evaluated by sulforhodamine B colorimetric assay. This assay enables the determination of the toxic concentration to 50% of the cells (CC_50_ value) by linking with the structural protein of viable cells. The technique provides crucial insights into the cytotoxicity of bioactive compounds and their potential use as therapeutics [25].

Each crude extract (*N. parvum* OQ30047 and *B. spartinae* OQ300436) was applied at different concentrations (500 μg/mL to 1.95 μg/mL) in every well in triplicate. After a period of incubation at 37 °C and 5% CO_2_ (48 h to VERO, 72 h to L929, and 96 h to A549) the plaques were fixed and colored with sulforhodamine B, and the absorbances were read using SpectraMax (Molecular Devices, San José, CA, USA) at 510 nm. The cytotoxic concentration to 50% of the cells, CC_50_, was calculated based on constructed concentration–response curves using GraphPad Prism 8.0 (Graph Pad Software 8.0.0 version, La Jolla, CA, USA).

#### 2.5.2. Virucidal Assay

Virucidal assays were conducted using Human Adenovirus type 2 (HAdV-2) as a model for non-enveloped viruses and Murine Hepatitis Viruses type 3 (MHV-3) as a model for enveloped viruses, following the DIN EN 14,476 protocol for the evaluation of the virucidal activity of chemical disinfectants and antiseptics in the medical field. The assay was performed in 96-well plates seeded with L929 and A549 cell lines, which are permissive to MHV-3 and HAdV-2 replication.

The objective of this assay was to challenge viral particles at different log dilutions (10^2^–10^6^) against compounds or extracts in suspension, with a contact time of 120 s, followed by an incubation period until visible cytopathic effects were observed (3–5 days for MHV-3 and 5–7 days for HAdV-2). This assay is crucial in evaluating the effectiveness of compounds or extracts as potential virucidal agents and may have significant implications in developing novel antiviral treatments.

#### 2.5.3. The Antiparasitic Assays Were Carried Out against Leishmania and Trypanosoma

For the screening of trypanocidal activity against *T. cruzi*, the Tulahuén strain transfected with the β-galactosidase gene was utilized in 96-well plates. Epimastigotes were treated with fungal total extracts at a concentration of 50 μg/mL, along with resazurin reagent at a concentration of 250 μg/mL. The resulting fluorescence was measured at 560 nm using a spectrophotometer. The positive control was benznidazole at a concentration of 10 μM, while the negative control was DMSO at 1%.

To conduct the initial screening, the *T. cruzi* Tulahuén strain transfected with the β-galactosidase gene [26] was cultivated for three days at 27 °C in LIT medium (Liver Infusion Tryptose) supplemented with 10% fetal bovine serum. The parasites were seeded at 0.54 × 10^6^ parasites/well and subsequently treated with fungal total extracts at a concentration of 50 μg/mL, diluted in LIT medium with 10% FBS and 1% dimethyl sulfoxide (DMSO–Merck^®^, São Paulo, Brazil). The positive control was the drug benznidazole (BZN, Sigma-Aldrich^®^) at a concentration of 10 μM (2.6 μg/mL), while the negative control was DMSO at 1%. The plates were incubated for 67 h in a 27 °C incubator. After treatment, 20 μL of resazurin (Sigma-Aldrich^®^) at a concentration of 250 μg/mL was added to all wells, and the plates were incubated for an additional 4 h in a 37 °C, 5% CO_2_ incubator. The fluorescence was measured at 560 nm using a spectrophotometer (Tecan^®^, Infinite M200, São Paulo, Brazil) with a reference wavelength of 590 nm [27].

Leishmania assay was performed using two strains of *Leishmania parasites*, namely *Leishmania* (L.) *infantum* strain MHOM/BR/74/PP75 and *Leishmania* (L.) *amazonensis* strain MHOM/BR/77/LTB0016. Parasites were added to microdilution plates at a concentration of 0.54 × 10^6^ parasites/well in M199 medium supplemented with 10% SBF and 2% human urine. Replicates of the parasites were treated with 20 μL of fungal extracts at a concentration of 50 μg/mL diluted in M199 medium with 1% DMSO. An amphotericin B positive control was included at 2 µM for *L. amazonensis* and 8 µM for *L. infantum*, while the negative control consisted of M199 medium with 1% DMSO. The plates were incubated for 72 h at 26.5 °C, followed by addition of 20 μL of a solution containing 250 μg/mL of resazurin solubilized in PBS buffer (pH 7.4) for 1 h and 30 min at 37 °C. Fluorescence was measured using a Tecan^®^ Infinite M200 microplate reader with excitation at 560 nm and emission at 590 nm. Cell viability was expressed as the percentage of mortality, calculated as the reduction in signal between the sample and experimental triplicate, divided by the average of the negative control minus one [28].

#### 2.5.4. Bacteriostatic Assay

The bacteriostatic assay was made using Luria broth sterile, 192 μL, 8 μL (0.6 OD) of bacteria *Escherichia coli* ATCC 11,303 and *Listeria monocytogenes* ATCC 19111, and 2 μL of each extract, ending at 500 μg/mL concentration into a 96-well plate, using as control wells with only broth and wells with broth and bacteria. The bacterial growth inhibition assay was conducted by incubating the bacterial culture for 12 h at 37 °C in a Spectramax microplate reader. During the incubation period, the optical density at 540 nm was measured every hour to determine the extent of bacterial growth inhibition [29].

#### 2.5.5. Yeast Assay

Species as *Candida albicans* ATCC 10231*, Candida krusei* ATCC 6258, *Candida parapsilosis* ATCC 22019, *Candida tropicalis*, and *Candida glabrata* (both clinical isolated) were cultured on Sabouraud dextrose agar for up to 18–48 h before the test. After this period, an inoculum was prepared using 5 distinct colonies (>1 mm in diameter) in 9 mL of water, which were then homogenized in a rotary shaker at approximately 2000 rpm. The cell density was adjusted to a 0.5 McFarland scale by measuring absorbance at 530 nm in a spectrophotometer, resulting in a concentration of 1–5 × 10^5^ CFU/mL. Subsequently, 90 μL of the suspension was inoculated into 96-well microplates and 10 μL of crude extracts at a concentration of 50 μg/mL, diluted in RPMI 1640 medium with 1% DMSO, were added. Amphotericin B 8 µM (Sigma-Aldrich^®^) was used as a positive control and RPMI 1640 with 1% DMSO as a negative control. The 96-well plates were incubated for 24 h in an oven at 35 °C. The plates were then read in a Tecan^®^ Infinite M200 microplate reader at an absorbance of 530 nm. Calculating the mortality rate consisted of the reduction between the sample blank and the experimental triplicate, divided by the mean of the negative control minus one [30].

## 3. Results

### 3.1. Chemical Profile of the Fungi-Derived Crude Extracts

The chemical profiles obtained by UPLC-ESI-HRMS analyses of *N. parvum* and *B. spartinae* crude extracts are provided in Table 1 and Table 2, respectively.

The chromatograms acquired in positive and negative modes revealed the presence of 16 compounds. The first metabolite at *t_R_* 0.42 min showed *m*/*z* 387.0875 corresponding to the molecular formula [C_23_H_16_O_6_-H]^−^. The precursor ion provided no fragmentation. However, a literature search suggested the structure of a polyketide derivative related to a bis-naphthoquinone [31]. Peak 3 showed its precursor at 1.15 with *m*/*z* 300.0878 corresponding to [C_16_H_15_NO_5_-H]^−^. No fragmentation was observed for this compound, and a literature review led to the structure of Pestauvicolactone A [32]. Peak 4 at 7.54 min with *m*/*z* 425.1372 [C_27_H_20_O_5_+H]^+^ was revealed to be a polyketide related to calanone, a coumarin identified from *Calophyllum* species [33]. The structure of peak 5 was assigned to be related to terpestacin, a fungus-derived sesterterpene [34].

Peak 6 showed its precursor at 8.35 min with *m*/*z* 415.2101 corresponding to the molecular formula [C_24_H_30_O_6_+H]^+^; no fragment was observed. The literature suggests a structure of a biphenyl produced by the marine fungi *Aspergillus* sp., identified as asperbiphenyl [35]. Peak 7 showed its precursor as sodium adduct [M + Na]^+^ at *t_R_* 9.08 min with *m*/*z* 467.2774 corresponding to [C_27_H_40_O_5_+H]^+^ and assigned as fusaproliferin, a fungal mycotoxin. The precursor led to fragments of *m*/*z* 427.2863 [M + H-H_2_O]^+^, indicated by a loss of 18 Da, followed by a loss of 60 Da indicated by *m*/*z* 367.2633 [M + H-H_2_O-COOCH_3_]^+^ [36]. Peak 8 at *t_R_* 9.59 min with *m*/*z* 530.3412 [C_33_H_45_NO_4_+H]^+^ was assigned as a sespendole, a fungal metabolite with an indolosesquiterpene core structure [37].

The precursor of *m*/*z* 544.3632 was assigned as peak 10 at 11.06 min corresponding to [C_32_H_49_NO_6_+H]^+^. By literature assessment, the metabolite was assigned as pestalotiopin B, a bioactive natural product with a sesquiterpenoid structure produced by endophytic fungi *Pestalotiopsis photiniae* [38].

Peak 11 was signed at 12.53 min with *m*/*z* 331.2851 corresponding to [C_19_H_38_O_4_+H]^+^; fragments with *m*/*z* 313.2743, 281.2473, 239.2385 indicate neutral loss of H_2_O (−18 Da), followed by methanol (−32 Da) and the neutral loss glycerol (−92 Da) at the precursor ion. The compound was identified as monopalmitin, and it has a role as a natural product found in plants and algae [39].

Peaks 12, 13, and 14 generated precursor ions in negative mode with *m*/*z* 279.2338, 255.2329, and 281.2493 at *t_R_* 12.60, 13.41, and 13.48 min corresponding to the molecular formula of [C_18_H_32_O_2_-H]^−^, [C_16_H_32_O_2_-H]^−^, and [C_18_H_34_O_2_-H]^−^. No fragmentation pattern was observed, and by literature review the metabolites were assigned as fatty acids compounds previously identified in fungi of the *Neofusicoccum* genus [40].

Peak 15 showed its precursor at 13.78 min with *m*/*z* 359.3150 corresponding to the molecular formula [C_21_H_42_O_4_+H]+. A fragment with *m*/*z* 341.3063 indicates neutral loss of H_2_O [M + H-18Da]^+^ from the glycerol moiety, and the metabolite is identified as monostearin. The literature reports the identification of fatty acid in plants [41], bacteria [42], and as a product of anaerobic fungus metabolism methane [43].

**Table 1 microorganisms-11-01599-t001:** Biocompounds extract of *N. parvum*.

Peak No	*t_R_* (min)	Identification	Molecular Formula	ESI(+) [M + H]^+^	ESI(−) [M − H]^−^	Fragment		References
				(*m*/*z*)	(*m*/*z*)	ESI(+)	ESI(−)	
1	0.42	Ethylidene-3,39-biplumbagin	C_22_H_16_O_6_		387.0875			[31]
2	0.79	NI	C_27_H_47_N_3_O_19_		716.2758			
3	1.15	Pestauvicolactone A	C_16_H_15_NO_5_		300.0878			[32]
4	7.54	calanone derivative	C_27_H_20_O_5_	425.1372	-		-	[44]
5	7.98	Terpestacin	C_25_H_38_O_4_	403.2865	-	385.2729	-	[45]
6	8.35	Asperbiphenyl	C_24_H_30_O_6_	415.2101	-	-	-	[46]
7	9.08	Fusaproliferin	C_27_H_40_O_5_	467.2774 ^a^	-	427.2863, 409.2721, 367.2633, 349.2526	-	[36]
8	9.59	Sespendole	C_33_H_45_NO_4_	520.3412	-	-	-	[38]
9	10.03	NI	C_31_H_45_NO_4_	496.3417	-	-	-	
10	11.06	Pestalotiopin B	C_32_H_49_NO_6_	544.3632	-	-	-	[47]
11	12.53	Monopalmitin	C_19_H_38_O_4_	331.2851		313.2743, 281.2473, 239.2385	-	
12	12.60	Linoleic acid	C_18_H_32_O_2_		279.2338			
13	13.41	Palmitic acid	C_16_H_32_O_2_		255.2329			
14	13.48	Oleic acid	C_18_H_34_O_2_		281.2493			[40]
15	13.78	Monostearin	C_21_H_42_O_4_	359.3150		341.3063	-	
16	14.58	-	C_24_H_38_O_4_	413.2662 ^a^		-	-	

Table 1 shows biocompounds identified using UPLC-ESIMS after solid-state fermentation of *Neofusicoccum parvum* using organic raw rice present in crude extract. ^a^ [M + Na]^+^.

**Table 2 microorganisms-11-01599-t002:** Biocompounds extract of *B. spartinae*.

Peak No	*t_R_* (min)	Identification	Molecular Formula	ESI(+) [M + H]^+^	ESI(−) [M − H]^−^	Fragment		References
				(*m*/*z*)	(*m*/*z*)	ESI(+)	ESI(−)	
1	0.42	Sucrose	C_12_H_21_O_11_	365.1072 [M + Na]^+^		325.1122, 205.0542		[48]
2	0.49	Citric acid	C_6_H_8_O_7_		191.0205			
3	1.37	2-Isopropylmalic acid	C_7_H_12_O_5_		175.0612			[49]
4	3.43	Phenylalanine	C_9_H_11_NO_2_		164.0339		120.0461	[50]
5	4.16	NI			245.1380			
6	4.38	Polyhydroxylated fatty acid	C_10_H_20_O_5_		203.1278 (−2.5)			
7	6.36	trihydroxy octadecenoic acid I	C_18_H_34_O_5_		329.2314 (4.3)		311.2223, 211.1303, 183.0106	[46]
8	6.73	trihydroxy octadecenoic acid I	C_18_H_34_O_5_		329.2314 (4.3)		311.2223, 211.1303, 183.0106	[46]
9	8.86	1-Myristoyl-2-lysophosphatidylcholine	C_22_H_46_N_1_O_7_P	468.3103 (2.8)		285.2425, 184.0734, 104.1070		[51]
10	9.59	1-Linoleoylphosphatidylcholine	C_26_H_50_N_1_O_7_P	520.3409 (1.2)		483.2483 [M + Na]^+^, 184.0734, 104.1070		[51]
11	10.11	1-Palmitoylphosphatidylcholine	C_24_H_50_N_1_O_7_P	496.3400 (−0.6)		313.2738, 184.0734, 104.1070		[51]
12	11.06	NI		358.3701				
13	12.60	Linoleic acid	C_18_H_22_O_2_		279.2325 (0.4)			
14	13.34	Palmitic acid	C_16_H_32_O_2_		255.2314 (−3.9)		-	[52]
15	13.48	Octadecenoid acid	C_18_H_34_O_2_		281.2477 (−1.4)			[53]

Table 2 shows biocompounds identified using UPLC-ESIMS after solid-state fermentation of *Buergenerula spartinae* using organic raw rice present in crude extract.

The chromatograms acquired in positive and negative modes for *Buergenerula spartinae* fermented extracts revealed the presence of 15 compounds. Peak 1 at *t_R_* 0.42 min showed a precursor *m*/*z* 365.1072 corresponding to the molecular formula [C_12_H_21_O_11_+Na]^+^, corresponding to sucrose. The precursor ion provided the ion with *m*/*z* 325.1122, indicating a loss of 18 Da [C_12_H_21_O_11_-H_2_O+H]^+^.

Peaks 2 and 3 showed precursors in negative mode at 0.49 and 1.37 min with *m*/*z* 191.0205 and 175.0612 corresponding to [C_6_H_8_O_7_-H]^−^ and [C_7_H_12_O_5_-H]^−^. No fragmentation was observed for these compounds, and a literature review indicated citric [54] and 2-isopropylmalic acids [55] as common natural products produced during fungi metabolism.

Peak 4 showed a precursor at 3.43 min with *m*/*z* 164.0339 corresponding to [C_9_H_11_N_1_O_2_-H]^−^, and mass spectra provided a fragment with *m*/*z* 120.0461, indicating a loss of 44 Da [C_9_H_11_N_1_O_2_-CO_2_-H]^−^, and the structure of phenylalanine was assigned as a metabolite present in the extract [56].

Peak 6 was characterized as a polyhydroxylated fatty acid based on the literature search [57]. The precursor produced no fragments.

Two peaks with precursor *m*/*z* 329.2314 at *t_R_* 6.36 and 6.73 min were assigned as peaks **7** and **8** corresponding to the molecular formula [C_18_H_34_O_5_-H]^−^. The mass spectra provided a fragment with *m*/*z* 311.2223 indicating neutral loss of 18 Da [C_18_H_34_O_5_-H_2_O-H]^−^, *m*/*z* 211.1303 corresponding to the fragment [C_12_H_19_O_3_]^−^, and literature assessment indicates isomers of long-chain fatty acid, identified as trihydroxy octadecenoic acid [58].

Three structures of phospholipids were assigned as peaks 9, 10, and 11 with *m*/*z* 468.3103, 520.3409, and 496.3400 corresponding to molecular formulas [C_22_H_46_N_1_O_7_P+H]^+^, C_26_H_50_N_1_O_7_P+H]^+^, and [C_24_H_50_N_1_O_7_P+H]^+^. The mass spectra provided two fragments that correspond to phosphatidylcholine ion *m*/*z* 184.0734 [C_5_H_15_N_1_O_4_P_1_]^+^ and choline ion *m*/*z* 104.1070 [C_5_H_14_N_1_O_1_]^+^ for all three compounds. Peak 9 provided one fragment corresponding to the cleavage of phosphatidylcholine fatty acid bonding with *m*/*z* 285.2425 that led to the identification 1-myristoyl-2-lysophosphatidylcholine. Peak 10 sodium adduct with *m*/*z* 483.2483 corresponded to 1-linoleoylphosphatidylcholine and peak 11 fatty acid ion *m*/*z* 313.2738 corresponded to 1-palmitoylphosphatidylcholine [51].

Peaks 13, 14, and 15 showed precursors in negative mode at *t_R_* 12.60, 13.34, and 13.48 min with *m*/*z* 279.2325, 255.2314, and 281.2477 corresponding to molecular formulas [C_18_H_22_O_2_-H]^−^, [C_16_H_32_O_2_-H]^−^, and [C_18_H_34_O_2_-H]^−^. No fragmentation was observed for the mentioned compounds, and the literature review indicated the fatty acids structures of linoleic acid [57], palmitic acid [59], and octadecenoic acid [53] as commonly produced by marine fungi.

### 3.2. Biological Assay

The sulforhodamine B assay showed low cytotoxicity (CC_50_ > 500 μg/mL) of extracts obtained from *N. parvum* and *B. spartinae* in cell lineages VERO, A549, and L929 as shown in Table 3.

Virucidal evaluation using crude extract of *N. parvum* and *B. spartinae* did not demonstrate activity against MHV-3 and HAdV-2 using DIN EN 14,476 protocols (no reduction of cytopathic effect observed).

A bacteriostatic assay using crude extracts against *Listeria monocytogenes* and *Escherichia coli* presented 98% growth inhibition in the first hour.

## 4. Discussion

The literature shows interactions as pathogens on tissue plants, saprophytes, and endophytic [60,61]. *B. spartinae* includes the family Magnaporthaceae, paraphyses hyaline, septate, dissolving at maturity, and conidiophores branched, being related as endophytic fungi, saprobic, and parasite [62].

*Neofusicoccum parvum* (Phylum Ascomycota) is a member of the Botryosphaeriaceae family and exhibits various types of interactions such as pathogenicity, saprophytic growth on decomposing matter, necrotrophic behavior by feeding on host cells, and endophytic associations, particularly in woody plants [60]. *Neofusicoccum parvum* has a worldwide distribution and is often associated with its pathogenic ability, specifically affecting stem regions, but it has also been identified as an endophyte [61].

The crude extract of *N. parvum* contained Ethylidene-3,39-biplumbagin, a quinone compound known for its bacteriostatic properties and ability to inhibit and reduce biofilm viability. In addition to these activities, quinones have also been shown to possess cardio-protective, anti-inflammatory, and analgesic properties. It is important to note that this compound was not previously associated with *N. parvum* or other endophytic fungi [63]. The bacteriostatic effects of quinones are attributed to their ability to reduce both gram-negative and gram-positive bacteria. Quinones are also known to inhibit the enzyme DNA gyrase, which is required for prokaryotic cell division, and are used in several antibiotics [64,65,66,67,68].

Calanone derivative is a coumarin compound first isolated from *Calophylum teysmannii*. It has been evaluated in cytotoxicity assays using HeLA and L1210 lineages, resulting in IC_50_ values of 22.8 and 59.09 μg/mL, respectively. An in vivo study using rats showed a reduction in breast tumors with a dose of 4 mg/mL [33,44,69]. These findings suggests that calanone derivative has the potential as an anticancer agent, but further studies are needed to explore its full range of biological activities. However, no literature exists on using this compound against bacteria or viruses.

Pestauvicolactone A was related as produced by the fungi *Pestalotiopsis uvicola* (CGMCC 3.8776); Hou and collaborators identified the molecule and evaluated against tumoral cells B16-BL6 (mouse melanoma), with no cytotoxic effect at 30 μM [31].

Sespendole is a sesquiterpene initially isolated from *Pseudobotrytis terrestris*, which has been evaluated for its antibacterial activity against *Bacillus subtilis* and *Mycobacterium smegmatis*, exhibiting a minimum inhibitory concentration (MIC) of 10 μg/6 mm per disk [70,71].

Pestalotiopinon B, a polyketide compound isolated from an endophytic fungi *Pestalotiopsis*, has been subjected to biological assays against bacteria and fungi. Still, both microorganisms were not inhibiting cell division [41]. These compounds have not been related to being produced by these fungi or other endophytic fungi in literature.

Terpestacin is a small molecule sesterterpenoid obtained first from the fungus *Arthrinium* sp. and produced by another endophytic fungus, mangrove *Fusarium proliferatum*.

Terpestacin is a sesterterpenoid, a small molecule obtained from the fungus *Arthrinium* sp. and produced by another endophytic fungus, the mangrove *Fusarium proliferatum*. Studies have shown that terpestacin possesses several biological activities, including angiogenesis inhibition and antifungal and antiviral (HIV) effects. Angiogenesis is a complex process involved in forming new blood vessels and growth. However, pathologic states can exploit this process to produce diseases such as tumor growth, metastasis, rheumatoid arthritis, and diabetes [34,72].

In addition to its angiogenesis inhibition activity, terpestacin has also shown antifungal activity against three different pathogen species, reducing hyphal growth, which may be attributed to its allelopathic action. These findings suggest that terpestacin may have potential therapeutic applications in treating various diseases [73].

Asperbiphenyl is a unique compound isolated from the marine fungus *Aspergillus* sp. that contains unsaturated fatty acid glycerol ester. It has been found to exhibit a significant inhibitory effect (35.5%) against the tobacco mosaic virus [35]. However, there is no literature evidence suggesting its efficacy against bacteria or other viruses.

Huaspenone C is a polyketide compound belonging to the N-containing furan-3(2H) group-one derivative. It was isolated from *Peyronellaeae* sp. and is the first naturally occurring N-bearing furanone derivative reported in nature. This study represents the second report of its kind [74].

Fusaproliferin, a sesterterpene initially identified from the culture of *Fusarium proliferatum*, has been the subject of numerous investigations, including cytotoxicity assays using SF-9 cells and IARC/LCL 171 [75,76]. Cytotoxicity using SF-9 cells and IARC/LCL 171 were evaluated, showing a CC_50_ of 100 μM and 60 μM [36]. The results showed a CC_50_ of 100 μM and 60 μM, respectively. Additionally, Cimmino (2016) demonstrated the compound’s activity against *Alternaria brassicicola*, *Botrytis cinerea*, and *Fusarium graminearum* [73]. However, there have been no reports of trials against bacterial and viral pathogens, making this study the first of its kind. The potential implications for future research in this area are vast and exciting.

The chemical characterization of the crude extract of *B. spartinae* led to the discovery of Hydroxyoctadecanedioic acid I, a fatty acid previously identified in the desert plant *Panicum turgidum* and, more recently, in Rosa damascena in 2020. Identifying this compound in *B. spartinae* opens up new possibilities for further research on its potential biological activities, such as its antioxidant, anti-inflammatory, or antimicrobial properties. The presence of this rare fatty acid in *B. spartinae* highlights the importance of investigating lesser-known species and their potential as sources of novel bioactive compounds [77,78].

Hydroxyoctadecanedioic acid has been previously identified in other plant species; its presence in the crude extract of *B. spartinae* is significant as it suggests the potential for unique secondary metabolites in this species. Further investigation and biological assays could reveal the potential therapeutic properties of these compounds, contributing to the development of new drugs and treatments. As such, identifying Hydroxyoctadecanedioic acid I in *B. spartinae* opens up a new avenue of research and discovery in natural product chemistry.

Phenylalanine is an essential amino acid that is found in a variety of foods and is also present in mother’s milk. Phenylalanine is structurally similar to dopamine, epinephrine (adrenaline), and tyrosine and can be converted into tyrosine, which in turn can be converted into catecholamine neurotransmitters. This suggests that phenylalanine supplementation may have antidepressant effects. Another compound metabolized from phenylalanine is phenylethylamine (PEA), which acts as a neurotransmitter and hormone and may serve as a neuromodulator for catecholamines. Studies have shown that PEA can increase extracellular levels of dopamine and modulate noradrenergic transmission. Additionally, PEA has been found to suppress the inhibitory effects of GABA(B) receptors. Therefore, like phenylalanine, supplementation with PEA has also been suggested to have antidepressant effects [79].

Several endophytic fungi have been found to produce phenylalanine, including *Cophinforma mamane*, *Penicillium citrinum*, and *Rhizopus oryzae* [80,81,82]. Moreover, L-phenylalanine dipeptide derivatives have been shown to have promising anti-cancer effects, particularly against prostate cancer cell lines PC3 and K562 cells in vitro [6].

2-Isopropylmalic acid (2-IPMA) is an essential intermediate in the biosynthesis of leucine in *Saccharomyces cerevisiae*. This compound is produced in the mitochondria from isoketovalerate and subsequently transported to the cytosol. Once in the cytosol, it undergoes two enzymatic steps to yield leucine [83]. Ricciutelli and collaborators evaluated the bacteriostatic capability of 2-IPMA present on wines against *E. coli* O157:H7, *S. aureus* ATCC 29213, *L. monocytogenes* ATCC 7644, and *S. enterica* ATCC 13314. The results indicate an MIC of 409 µg/mL [84]. Being correlated with inhibition was observed in this research.

1-Linoleoylphosphatidylcholine was identified by Zhang and collaborators (2010) as a biomarker present on plasma of patients with liver failure caused by hepatitis B virus, with no biological activity demonstrated in the literature or production by endophytic fungi [85].

1-Palmitoylphosphatidylcholine was identified to be produced by *Perna canaliculus* and evaluated as anti-histaminic, but with no evidence of activity against pathogens [86].

Interestingly, *Neofusicoccum parvum* has been found to produce a more significant number of bioactive compounds compared to *B. spartinae*. These fungi are commonly associated with mangrove plants, and further investigation into their bioactive potential may yield promising results for biocontrol and other applications.

## 5. Conclusions

Exploring endophytic fungi from mangroves has revealed many bioactive compounds with potential applications in various fields, including medicine, agriculture, and biotechnology. Our investigation into *N. parvum* and *B. spartinae* has demonstrated their ability to produce bioactive compounds that exhibit low cytotoxicity and inhibit bacterial growth.

While no inhibition was observed against model viruses, yeasts, Trypanosoma, and leishmania, the potential application of these crude extracts against other pathogens warrants further investigation. Overall, our findings highlight the potential of endophytic fungi from mangroves as a promising source of novel bioactive compounds with various applications.

## Figures and Tables

**Table 3 microorganisms-11-01599-t003:** Cytotoxic results.

Extracts	L929	A549	VERO
*Neofusicoccum parvum*	>500 μg/mL	>500 μg/mL	>500 μg/mL
*Buergenerula spartinae*	>500 μg/mL	>500 μg/mL	>500 μg/mL

Table 3 shows CC_50_ of crude extracts evaluated on different cell lineages.

## Data Availability

The data presented in this study are openly available in GenBank at https://www.ncbi.nlm.nih.gov/nuccore/OQ300478 (accessed on 23 January 2023), reference number OQ300478.1. The data presented in this study are openly available in GenBank at https://www.ncbi.nlm.nih.gov/nuccore/OQ300436 (accessed on 23 January 2023), reference number OQ300436.1.

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
