# Peer review of "Bioactivity Screening and Chemical Characterization of Biocompound from Endophytic Neofusicoccum parvum and Buergenerula spartinae Isolated from Mangrove Ecosystem"

_microorganisms, 2023, doi:10.3390/microorganisms11061599_

Round 1

Reviewer 1 Report

The paper is well structured and the experimental delineation and description is concise to understand all manuscript. Results are properly presented and discussed. 

-       Some typo correction were found in the manuscript related with the symbol of temperature used., please see carefully section 2.3., second paraph.  

-       Check word concentrantion in section 2.5.5.

In tables 1 and 2, condition of the obtained data should be provided in the description of the table, as to be understandable individually apart from the manuscript. ( Some of dexcription put as Table information , should be in the caption).

Supplementary material should be provided in appropriate form, with figure caption.  

Author Response

#Revisor 1 Comments and Suggestions for Authors

The paper is well structured and the experimental delineation and description is concise to understand all manuscript. Results are properly presented and discussed. - Some typo corrections were found in the manuscript related with the symbol of temperature used., please see carefully section 2.3., second paraph.
Response: Thank you for the observation, symbol of temperature was adjusted.

- Check word concentration in section 2.5.5.
Response: The word was corrected, thank you for the observation on section 2.5.5 (concentration).

In tables 1 and 2, condition of the obtained data should be provided in the description of the table, as to be understandable individually apart from the manuscript. (Some of description put as Table information, should be in the caption).
Response: Thank you for the suggestion. Both tables received captions with better description.

Supplementary material should be provided in appropriate form, with figure caption.
Response: Thank you for the suggestion, figures captions were included.

Reviewer 2 Report

the tudy was well-planned, and done. Data is clearly presented.

I feel the introduction is too long. Better to remove the first part.

In M & M, author must mention the way they isolated the strains and the way they determined the species.

Language is good.

Author Response

#Revisor 2

Comments and Suggestions for Authors

The study was well-planned, and done. Data is clearly presented.

Response: Thank you very much.

I feel the introduction is too long. Better to remove the first part.

Response: Thank you for the suggestion. First part was removed.

In M & M, author must mention the way they isolated the strains and the way they determined the species.

Response: Thank you for the suggestion, isolation and determination process were included on section 2.1 page number 3.

“Disinfection was made at the University Federal of State of Santa Catarina according to protocol, using a rinse of ethanol, hypochlorite, and water; following the process of isolation, internal parts of samples required to be cut and exposed internal factors at potato-agar-dextrose (Kasvi) in Petri dishes and maintained incubated for 14 days at 28°C [20].

The identification of the endophytic fungi isolated from leaves of A. schaueriana and Laguncularia racemosa demanded a thorough analysis of micromorphological characteristics, including the distribution of fragments and microculture, as well as macromorphological and molecular analysis of the ITS1 region of rDNA [21]”